# Barriers and facilitators of fetal heart monitoring with a mobile cardiotocograph (iCTG) device in underserved settings: An exploratory qualitative study from Tanzania

**Dorkasi L. Mwakawanga**[1,2], **Sanmei Chen**[1], **Yhuko Ogata**[3], **Minami Suzuki**[4], **Yuryon Kobayashi**[5], **Miyuki Toda**[1], **Naoki Hirose**[1], **Yoko Shimpuku**[1] *

**1** Graduate School of Biomedical and Health Sciences, Hiroshima University, Hiroshima, Japan, **2** Department of Community Health Nursing, Muhimbili University of Health and Allied Sciences, Dar es Salaam, Tanzania, **3** Melody International Ltd., Kagawa, Japan, **4** Castalia, Co. Ltd., Tokyo, Japan, **5** Translational Research Center Biodesign Team, The University of Tokyo Hospital, Tokyo, Japan

* yokoshim@hiroshima-u.ac.jp

## Abstract

### Background

Fetal monitoring in low-resource settings is often inadequate. A mobile cardiotocograph fetal monitoring device is a digital innovation that could ensure the safety of pregnant women at high risk and their fetuses through early detection and management of fetal distress. Research is scarce on factors that affect the implementation of fetal heart monitoring using the mobile cardiotocograph device in low-resource settings, including Tanzania. This study aimed to explore the barriers and facilitators of fetal monitoring with a mobile cardiotocograph device in Tanzania.

### Methods

We adopted an exploratory qualitative study to analyze the barriers and facilitators of fetal monitoring using the mobile cardiotocograph device in primary healthcare facilities. Seventeen face-to-face in-depth interviews with healthcare providers and seven focus group discussions with women were conducted. Braun and Clarke's thematic analysis guided the data analysis. It included the following steps: familiarizing with data, generating initial codes, searching for themes, reviewing themes, defining and naming themes, and producing the report.

### Results

Three themes emerged as barriers: individual-related ones, including inadequate knowledge and skills to use mobile cardiotocograph devices, institutional barriers attributed to limited referral infrastructures and staff shortage, and community-related barriers, such as myths and misconceptions that limit antenatal care checkups. Individual factors, including motives and desires of healthcare providers and community trust in the healthcare system,

**Data Availability Statement:** "All relevant data are within the paper and its Supporting Information files."

**Funding:** This study was funded by Japan Agency for Medical Research and Development (AMED) under Grant Number JP22hk030212.

support systems related to training and mentorship opportunities for healthcare providers, and the availability of community-based health programs in the respective areas, were revealed as facilitators of mobile cardiotocograph devices.

## Conclusion

Implementing iCTG in primary healthcare facilities is affected by several factors, from individual to institutional level. Providing user-friendly procedure manuals and training on the functions of the iCTG device and cardiotocograph interpretations could serve as potential solutions to improve the competence and confidence of healthcare providers. Moreover, the availability of supportive infrastructure, adequate human resources for health, and community sensitization were good points to start with when addressing institutional and community barriers. Nevertheless, multistakeholder engagement during the design and implementation of fetal monitoring using a mobile cardiotocograph device is warranted.

## 1. Introduction

Low and Middle-Income countries (LMICs) experience a high burden of maternal and newborn mortality. Globally, an estimated 2.4 million newborns died in 2020 [1]. Almost all of these deaths (99%) occur in LMICs, with Sub-Saharan Africa accounting for 66.3% [2–4]. In Tanzania, the neonatal mortality is at 24 deaths per 1000 live births [5], which is much higher than the one indicated in SDG 3. Prematurity, infections, and perinatal hypoxia are the leading causes of neonatal deaths in these countries [6]. The best outcomes occur when these conditions are prevented rather than treated. Intensive monitoring of the fetal heart rate (FHR) is essential for early detection and diagnosis of perinatal hypoxia and to prevent stillbirth and neonatal mortality resulting from asphyxia [7]. Many medical devices designed for use in high-income countries have several limitations when reciprocated in low-resource settings [8, 9]. Currently, technological innovations are being leveraged to ensure the availability of suitable and appropriate devices for monitoring fetal surveillance in LMICs [8].

A cardiotocograph (CTG) has been widely used in high-resource countries for FHR monitoring during late pregnancy and around childbirth to allow the early detection of abnormal FHR and the prevention of complications resulting from fetal hypoxia [10]. Detecting a hypoxic state early is, therefore, the first step in potentially preventing fetal hypoxia, which may result in stillbirths and newborn deaths [11]. According to studies, approximately 2.6 million stillbirths and 4.2 million newborn deaths could be prevented by high coverage of life-saving interventions combined with equipping health facilities with the proper equipment and training healthcare providers [12]. Several barriers limit the use of CTG in limited-resource countries, including high costs, limited competencies among healthcare workers, and a lack of continuous electricity supply.

Recently, technological innovations created various mobile FHR monitoring devices that could ensure the safety of pregnant women at high risk and their fetuses in underserved settings with medical resource challenges, including a shortage of staff [9]. One of these devices is a mobile cardiotocograph device (iCTG). The iCTG (Melody International Ltd) includes a wireless Doppler FHR monitor, a wireless uterine contraction monitor, and a tablet [13]. When adequately charged, the iCTG device provides access to continuous FHR monitoring, stores data obtained on the cloud server, and transmits it via the internet [13]. Therefore, the

FHR pattern can be assessed on-site or remotely by medical personnel using a tablet or laptop and aid healthcare providers in the early detection of perinatal hypoxia and prompt caesarean section [14]. The introduction of iCTG in Thailand has increased the rate of antenatal check-ups for mothers with high-risk pregnancies, which has increased the chance of survival for both the expectant mother and the fetus [13, 15].

In Tanzania, as in other LMICs, auscultation using a fetoscope and occasionally using a fetal doppler, coupled with the use of partogram documentation, remains the most common method for monitoring fetal heartbeats during pregnancy and labour, notwithstanding many challenges like the severe shortage of skilled human resources for health [16–19]. The previous use of an FHR monitor called Moyo in Tanzania demonstrated a breakthrough in improving perinatal survival by identifying fetuses at high risk of intrapartum hypoxia-ischemia [20, 21]. Moreover, qualitative findings among mothers showed that it was a preferred device due to the mothers'ability to hear the fetal heart sounds, which provided reassurance of their fetus's health condition [22]. Similarly, the iCTG device is user-friendly, facilitates early detection and prompt intervention, and enables monitoring of foetal conditions. At the same time, a woman is being transferred, thereby increasing the likelihood of saving a life. However, following the implementation of Moyo, healthcare providers were still unsure when to use Moyo and how to differentiate between maternal and fetal heart rates [23]. The latter indicates the need to understand further the barriers and facilitators to implement innovative technologies for FHR monitoring in underserved countries, including Tanzania.

Therefore, we conducted exploratory qualitative research to identify the barriers and facilitators for implementing fetal heart monitoring with an iCTG device at primary healthcare facilities in the Pwani region of Tanzania. This was done as part of piloting an implementation study to examine the impact of iCTG on pregnancy and childbirth outcomes. It is critical to conduct qualitative research before implementing evidence-based health interventions to gain insights into the different perspectives of stakeholders and identify potential barriers and facilitators to adopting new practices. Qualitative methods answer complex questions such as how and why efforts to implement best practices may succeed or fail and how consumers and providers experience and make decisions in care [24]. In addition, it helps researchers to understand contextual factors, uncover hidden needs, and align with context-specific implementation plans that address both practice issues, the knowledge and skill needs of providers, and the attitudes of consumers toward adopting a new health behaviour, ultimately leading to more informed and effective implementation strategies. Therefore, evidence for considering implementation barriers and facilitators is needed to determine the optimal fetal monitoring strategies in low-resource countries.

## 2. Methods

### 2.1 Study design and context

We employed an exploratory research design using in-depth interviews (IDIs) and focus group discussions (FGDs) to examine the facilitators, barriers and strategies for iCTG implementation in two selected districts of the Coast region in eastern Tanzania. The study was conducted in Kisarawe and Bagamoyo districts, including two hospitals and five health centers. Despite both the district hospital and the health center falling within the primary level of the pyramidal Tanzanian health service delivery system, the district hospital functions as a referral hospital for their immediate health facilities due to their greater capacity to provide comprehensive emergency obstetrics and newborn care. Both facilities have a role of providing prenatal care as one of their core function includes monitoring the fetus and woman's condition. Consequently, the two districts were purposefully chosen due to the similarities between their

rural and semi-urban populations, with the majority living in rural settings with limited access to quality prenatal care. The latter could provide a better picture of implementation barriers and the opportunities towards fetal monitoring implementation with iCTG in these settings to enhance perinatal safety.

## 2.2 Study participants and recruitment

The recruitment of study participants was conducted face-to-face between November 2022 and June 2023. This study involved district reproductive and child health coordinators, doctors, midwives, pregnant women and postnatal mothers (**see Table 1**). We used a purposive sampling technique to recruit HCPs from different cadres and years of working experience to get rich and diverse perspectives. We involved HCPs of various levels, those responsible for supervision and training (clinical services coordinator and district RCH coordinator) and those involved in providing prenatal or delivery care. Pregnant women and postnatal mothers were recruited purposefully from the antenatal and postnatal ward or clinic based on age, education and parity differences. The researcher identified the participants with the assistance of the facility in charge and the district reproductive and child health coordinator. Each identified participant was approached face-to-face by the researchers, who explained the purpose of the contact and set up an appointment for an interview. For FGDs, pregnant women and postnatal mothers present at the particular facility during the data collection period were first approached by their immediate midwife on duty, who introduced the researchers to them. Then, the researchers selected participants who met the inclusion criteria, explained the purpose of the study, and requested that they provide their consent to participate in the FGDs. We recruited 17 HCPs, 25 postnatal mothers and 35 pregnant women from both districts.

## 2.3 Patient and public involvement

Patients and the public were not directly involved in the design of this study. However, they were involved in the workshops on disseminating the findings and program implementation plans in their respective catchment areas.

## 2.4 Ethical approval and consent to participate

The study was carried out following the relevant guidelines and regulations. Ethical approval was obtained from the National Institute for Medical Research, Tanzania (NIMR/HQ/R.8b/Vol.I/1097) and the Ethical Committee for Clinical Research of Hiroshima University, Japan (C2021-0340). The permission to conduct this study was obtained from the respective district medical authorities and the healthcare facility in charge. To protect study participants' privacy and personal information, confidentiality and anonymity principles were ensured by referring to them using numbers during data collection. After receiving information regarding the aim

**Table 1. Study participants (In-depth interviewee and focus group discussants).**

| Participants | Number of participants | Data collection method | Number of data collection |
|---|---|---|---|
| Doctors | 8 | IDI | 8 |
| Midwives | 7 | IDI | 7 |
| District RCH Coordinator | 1 | IDI | 1 |
| Clinical Services Coordinator | 1 | IDI | 1 |
| Pregnant women | 35 | FGD | 4 |
| Postnatal mothers | 25 | FGD | 3 |

and the consent to be audio recorded, we obtained written informed consent from each study participant. Participants were told that their participation in the study was voluntary and that they may withdraw at any time.

## 2.5 Data collection

We developed semi-structured interviews and FGD guides in English and translated them into Kiswahili (see **S1 File**). We used both languages in data collection, where the English guide was used to collect data from HCPs while the Kiswahili guide was used to conduct FGDs with women. The guides were prepared based on the experience of the research team with the Tanzanian health system and the relevant literature on the barriers to and strategies for implementing evidence-based interventions in LMICs [25, 26]. We pretested the guides to check the clarity of questions, to see if the participants understood them well, and to see if they captured what we wanted to capture. The data collection was conducted with three members of the research team (DLM, MS, YK), of whom one had a bachelor's degree and two had a master's education level. Before data collection, the data collection members and the research assistants (RAs) reviewed the study objectives, guides, informed consent, and research procedures. The data collection members conducted the interviews and the FGDs while the research assistants took notes and audio-recorded the sessions. All audio-recorded files were stored in a password-protected folder on a researcher's computer and were shared with other researchers for analysis. We used 25 days to conduct 17 interviews and 7 FGDs, whereby one interview or FGD was carried out in a day, followed by a debriefing meeting among the data collection team to hear what transpired during the interview or FGD. Saturation was used to determine the number of interviews and FGDs (i.e., we stopped data collection when no new information was gathered from participants and redundancy was reached) [27].

**In-depth interviews.**   We conducted 17 IDIs with 2 CHMT members, eight medical doctors, and seven midwives. The interviews were conducted in English, and all interviewee were asked beforehand about their comfort level with the language. The questions in the interview guide focused on the existing systems for providing prenatal care and the needs, barriers and facilitators for implementing fetal heart monitoring using a new mobile cardiotocograph device (iCTG). The interviews were conducted at a participant's convenience in a secure, private, and quiet room with fewer distractions on the hospital premises. Each interview was audio-recorded. The interviews lasted between 45 and 60 minutes.

**Focus group discussions.**   Seven FGDs were conducted, 4 with pregnant women and the other 3 with postnatal mothers in Kiswahili language. Each group consisted of 8 to 10 participants. All FGDs were conducted by one researcher (DLM) with experience in qualitative data collection and one RA who took field notes. The midwife in charge of the antenatal clinic or postnatal ward assisted researchers in organizing and choosing a private and quiet place to conduct FGDs. All FGD responses were audio-recorded, and field notes were taken. Before each FGD, participants provided written informed consent after being informed of the purpose of the study and that the session would be audio recorded. FGDs lasted between 60 to 110 minutes.

## 2.6 Data analysis

First, the audio-recorded interviews and discussions were transcribed verbatim. Data were analyzed using Braun and Clarke's thematic analysis approach [28]. Before doing the analysis, four researchers read and re-read transcripts and field notes to familiarize themselves with the data and context and gain a general understanding of the participants' perspectives. The analysis team included a midwife specialist, a qualitative researcher, a biodesign expert and a nurse.

The women's FGD transcripts were analysed in Kiswahili by the primary author (DLM), a native speaker with significant background knowledge of the context and experience in qualitative research. The interpretation was then conducted iteratively across the textual materials in both languages, with a write-up in English to facilitate collaboration between international team members. The codebook with initial codes was prepared deductively from preexisting ideas and inductively from emergent information observed during data familiarization. The data was managed and organized using Dedoose software. The open coding was done in pairs to ensure inter-coder reliability. Before coding more transcripts, we convened to resolve the discrepancies and disagreements that emerged after coding the same transcript. The codebook was updated to include all new emergent codes as the analysis progressed. The identified codes were then examined for similarities, and subthemes were developed. The subthemes were compiled, and recurring patterns across the data set were identified as themes. Themes were refined and finalized through review and discussion with the entire team of researchers. The generated sub-themes and themes were assigned to their corresponding pre-determined main group, representing either a barrier, a facilitator or a strategy for iCTG implementation. Finally, all findings were presented with quotes describing each theme's meaning.

## 2.7 Trustworthiness

The four criteria proposed by Lincoln and Guba [29] were used to ensure the trustworthiness of the study's findings. These include credibility, dependability, confirmability and transferability. The credibility was enhanced through member checks and the triangulation of study participants (providers and clients). Member check was conducted with participants of the IDIs as research members travelled to the sites twice and asked directly if our understanding of the interviews represented what they answered during the interviews. The triangulation of data collection methods, study settings, and researchers enhanced the credibility and dependability of this study. Furthermore, the study involved audio-recording of interviews and discussions to capture participants' accounts, with transcripts produced from multiple sources to ensure data credibility through maximum use of participant accounts and numerous data collectors. To confirm that the findings reflected the participants' perspectives and not the researchers' understanding of the research topic, themes were generated inductively using thematic analysis and presented with the support of subthemes and succinct quotes. Further, FGDs were conducted by DLM, and as one of the midwife specialists with an adequate understanding of the research topic, it is likely to have influenced the interpretation of data. Nevertheless, data analysis was done jointly with the sixth author, and the team reached a collective decision on the naming themes with multiple disciplines. The transferability of this study's findings is enhanced by describing the study setting, context, data collection methods, process, and analysis [30].

# 3. Results

## 3.1 Demographic characteristics of study participants

Pregnant women and postnatal mothers ranged from 18 to 41 years. Out of 60 FGD participants, 25 were postnatal mothers, of whom nine had given birth once, and 16 had given birth twice or more. Thirty-five were pregnant women, of which ten were pregnant for the first time. Out of 60 women, 30(50%) completed primary school, 21(35%) had completed secondary school, 5(8.3%) had attended college and 4(6.7%) had no formal education. About half of the women, 30(50%), work in tiny trades, followed by 17(28.3%) who were housewives and 13 (21.7%) who were either farmers or tailors.

Regarding participants of IDI, two of the nine doctors were medical specialists (pediatricians and gynecologists), and seven were medical registrars. Four of the seven midwives had diplomas in nursing, and three had bachelor's degrees in nursing or midwifery. Healthcare providers averaged 2 to 8 years of working in the maternity department.

### 3.2 Sub-themes and themes emerged from thematic analysis

From the analysis of IDI and FGDs, 62 codes were generated, which formed 12 sub-themes that resulted in three themes of barriers and three themes of facilitators for implementing a mobile fetal heart monitor iCTG (see **Fig 1**). The barriers were grouped as 1) individual-related barriers that include inadequate knowledge and experience about iCTG and language barrier, 2) institutional-related barriers that include inadequate and poor infrastructures for referrals, unfriendly referral communication platforms, and inadequacy of human and non-human resources, and 3) community-related barriers that include illiteracy regarding ANC checkup and myth and misconceptions surrounding childbirth. The facilitators included: 1) Individual factors such as motives and desire of healthcare providers to use the iCTG for FHR monitoring and community trust in the health system, and 2) availability of support systems to improve care in primary health facilities. **Table 2** shows the potential solutions mentioned by the participants.

**Barriers.** *Individual related barriers.* **a) Inadequate knowledge and experience about iCTG.** Most healthcare providers interviewed in this study stated that they had never seen or used a CTG. Despite not understanding how to use it, some claimed to have seen it in private hospitals. While practically all participants reported being unaware of and having never seen the mobile fetal heartbeat monitor iCTG, one participant said, *"CTG? No, I don't know it . . . we don't use it because we don't have . . . ooh, we use a fetal scope to check fetal heart rate* **(Midwife at a health centre 5)**

**b) Language barrier.** Some participants expressed that using English in the instructions manual for medical equipment introduced into healthcare facilities has prevented healthcare

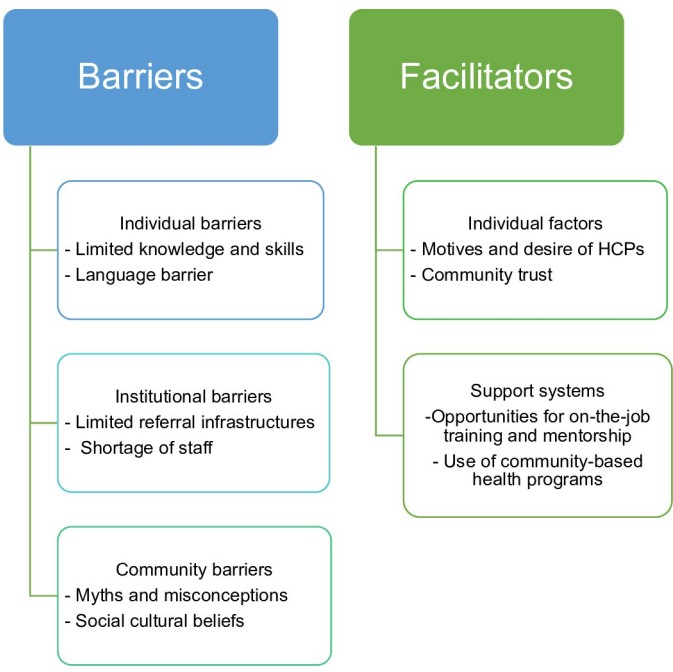

**Fig 1. Summary of findings.**

**Table 2. Potential solutions to the identified barriers.**

| Barriers | Potential solutions |
|---|---|
| Competence gap | Capacity building for healthcare providers<br>• Mentorship<br>• On-the-job training |
| Language barrier | Provision of language user-friendly procedure manuals |
| Inadequate and unfriendly referral system | Stationing the ambulances in areas/zones with referral challenges |
| Shortage of staff | Ensure adequate human resources for health<br>• Task sharing<br>• Use of volunteer schemes to higher through council funds<br>• Request recruitment from the central government |
| Illiteracy regarding ANC checkup | Provision of health education in healthcare facilities and community outreach |
| Community myths and misconceptions | Raising awareness through health education |
| Less ANC attendance/defaulters | Use of community health workers for community sensitization |

providers from operating the equipment. Participants regarded this as a barrier to some individuals who may be more confident and comfortable reading the instructions in Kiswahili rather than English.

> *"Sometimes the equipment is brought here, but the manuals to operate are written in English and not everyone could understand everything written in English, we end up not using it . . . some do not have any instructions"* **(Doctor at a health centre 3)**

*Institutional barriers.* **a) Inadequate and poor infrastructures for referrals.** Participants expressed their concerns about the rigidity of the referral system. They claimed that because health centers lack standby ambulances, they must call the district hospital to arrange an ambulance to pick up the patient. They also stated that the ambulance wasn't always available owing to fuel shortages or maintenance-related issues with the vehicle. The existing ambulances are not equipped sufficiently to assist with patient care throughout transportation. Participants also raised concerns about the long distance between a health center and a district hospital, as well as how bad roads put patients at risk of dying while waiting for an ambulance or being transferred.

> *"Sometimes we fail to save a life because we don't have the equipment to manage the case, we need to transfer the patient and the distance is almost 40km to a district hospital. . .no standby ambulance at the health centre, an ambulance has to come from this distance to take the patient in total is almost 90km so can we save life in this situation?"* **(Doctor at a health centre 3)**

**b) Unfriendly referral communication platforms.** Participants reported no appropriate channels for sharing referral-related information about the woman being transferred. They said that whenever a healthcare provider at a district hospital or health centre recognizes a patient who requires a transfer to a higher-level hospital, they should record the patient's details and the reason for the referral on a formal referral letter, which was paper-based. This was irrespective of their wish to have an online platform for sharing patient data with the healthcare providers at the recipient healthcare facility. Additionally, they had to use a personal phone to call a healthcare provider at the recipient facility before transferring.

*"If we want to make a referral, we write the details of the patient on a referral letter, but we also have to call the hospital where we take the patient to inform them; sometimes we use our phones to share the patient information* **(Doctor at a health centre 4).**

**c) Inadequacy of human and non-human resources.** Participants informed us that while there is an upsurge in the number of clients daily, thus increasing the workload, the number of skilled providers has been relatively small. They expressed that most primary-level healthcare facilities only had two or fewer staff to provide antenatal and delivery services. Some participants remarked that providing maternity care has been challenging due to inadequate equipment and protocols.

*"Because of less staff, always a midwife conduct delivery and stays with patients all the time and consults a doctor when she wants to share information or when she wants assistance to manage a patient. . .per shit there is one midwife and one doctor"* **(A midwife at a health centre 3)**

*Community-related barriers.* **a) Illiteracy regarding ANC checkup.** Pregnant women in the focus group admitted that they occasionally skip their follow-up checkup visits without a valid reason. Some participants had lower ANC attendance and provided valid explanations for their lower ANC attendance frequency. One participant said that when she arrived at the facility and saw that many women were waiting for services, she fled and returned home without getting checked out.

*"When I come here (health facility), I wait up to 5 hours, sometimes 6 hours, to finish everything, but there is no way we want to get services . . . sometimes if I come and find a very long queue, I opt to go home without a checkup and come back next time"* **(FGD of Pregnant women at a district hospital A)**

**b) Myths and misconceptions surrounding childbirth.** Healthcare providers claimed that some women are ignorant and that others fear coming to the health facility for a checkup. Some were afraid of being diagnosed with a danger sign that might predispose them to have a referral. Others who were informed to undergo a Caesarean section because of their medical or obstetrical complication couldn't accept it readily and believed they could still give birth normally. Other women delayed starting their prenatal care because they thought that if they did, evil people might notice them and cause miscarriages.

*"Some women are ignorant of risks, those who are high-risk and are indicated to deliver through C-section because it may be dangerous to subject them to a normal delivery, some women do not accept . . . link with myth and beliefs about reasons for C/S. . .they insist on trying the normal delivery even when is not possible"* **(Midwife at a health centre 4)**

**Facilitators.** *Motives and desire to use iCTG for FHR monitoring.* **a) Awareness about continuous FHR monitoring.** Some participants, particularly those with a specialized education, agreed they had learned about continuous FHR monitoring in school. They also emphasized the benefits of using the Moyo Doppler, which they had previously observed to be simple to use and move around, identify fetal wellbeing, and facilitate prompt decisions to avert complications associated with hypoxia, like birth asphyxia.

*"The device is excellent and can screen fetal condition. It is easy to use and can identify those with fetal distress when compared to using a fetal scope to listen for abnormal fetal heartbeats*

*if we had one, it might help prevent cases of birth asphyxia"* **(Obstetrician at a district hospital B)**

**b) Need and readiness to receive training and use iCTG.** Participants expressed a desire for and interest in using a reliable FHR monitor, such as a Doppler and a portable CTG. Additionally, they expressed a strong desire to learn how to use these devices properly. Participants acknowledged that even though they occasionally receive similar equipment like portable ultrasound, operating it without training or instructions can be challenging.

*"We need to receive training on how to operate this equipment (iCTG) because sometimes there was equipment brought as donations, but we didn't have anyone to give instructions on how to use them, that has been a problem"* **(Obstetrician at a district hospital A)**

The clinical coordinator at a district hospital said that a trustworthy tool is required to confirm the fetal status to make an early decision to perform a caesarean section to save the baby's life and avoid any delays that cause intrauterine fetal deaths.

*"There was a time when we had a woman who lost a baby because we were unsure of the fetal heart rate. It took us a long time to decide to perform a Caesarean section because we lacked a reliable device to confirm the condition of the fetus. When we went through with the procedure, there had already been an intrauterine death . . . we saved the mother, but we lost the baby, which is a terrible situation"* **(Clinical coordinator at a district hospital B)**

*Availability of support systems to improve care in primary health facilities.* **a) On-the-job training and mentorship.** Participants noted receiving mentorship from the council and the higher-level health management team. They also reported getting comprehensive emergency obstetric and newborn care (CEmONC) from the Ministry of Health and doing intra-facility training.

*"We receive CEmONC training on life-saving skills, we also collaborate between health centres and a district hospital to conduct various trainings among ourselves to get updated skills in managing obstetric emergencies"* **(Doctor at a health centre 2)**

**b) Use of community health workers.** Participants from both research site districts reported using community health workers (CHWs) to identify pregnant women in their captive communities, follow up with them to make sure they attended prenatal care as required, and occasionally escort them to the hospital for delivery.

*"We use community health workers in this area to remind women to attend their follow-up checkups and when to use the services of a health facility, as well as to educate the community and women on the value of antenatal checkups"* **(Midwife at a district hospital A)**

*Community trust in the healthcare system.* **a) Receiving ANC services from healthcare facilities.** Pregnant women admitted during the focus group discussion that they visit medical facilities for prenatal checkups primarily to know about the growth and health of the fetus within the womb. Additionally, they acknowledged that they had learnt about birth preparation from healthcare providers, and thus, their understanding was appropriate. They also expressed a strong desire to hear their fetal heartbeat sounds when asked if they would like to do so when they came in for their appointment.

*"I visit the hospital to know how my unborn baby is doing inside the womb. Remaining unchecked while I am aware that there might be changes is not good. . . I am interested in how things are going, whether his (fetus) movements are fine and whether his (fetus) heartbeats are normal"* **(FGD of Pregnant women at health centre 1)**

## 4. Discussion

We aimed to analyze the barriers and facilitators to implementing iCTG, specifically in primary healthcare facilities, to inform the development of intervention and training for healthcare providers. To the best of our knowledge, this is among the few qualitative studies that explored the barriers and facilitators of implementing iCTG devices in primary healthcare settings in Tanzania. Our findings revealed varying experiences that reflect barriers and facilitators for implementing the iCTG in primary healthcare facilities. The barriers were highlighted as individual (inadequate competencies), institutional (inadequacy of referral infrastructures and shortage of staff) and community-related (illiteracy about maternal care services, myths and misconceptions). The existence of support systems to improve care in primary healthcare facilities, the motives and desire of healthcare providers to improve care, and the community's trust in health systems were significant facilitators revealed by this study. The barriers and facilitators reported in our study are consistent with findings from other studies conducted in settings similar to Tanzania's regarding the feasibility of introducing new technology into the practice [31–33].

The adequacy of the knowledge and skills of healthcare providers is an essential component of using and interpreting the CTG data accurately. As found in this study, inadequate knowledge among healthcare providers poses a barrier to the efficient and effective implementation of iCTG. Because the CTG is almost unavailable in Tanzanian health facilities, health students receive little or no structured formal training on its use for fetal heart rate and labour monitoring. As a result, the majority were unaware and had no experience using continuous fetal monitoring, either the regular CTG or the mobile CTG. This is consistent with what was documented in other studies where a lack of knowledge and interpretive skills limited the effectiveness of CTG [34, 35].

Nevertheless, as found in our study, the availability of opportunities to provide in-service training and supportive supervision serve as possible solutions to fill the knowledge and skills gap. A growing body of evidence shows that in-service training benefits healthcare providers' understanding of CTG practice [36, 37]. Moreover, providing user-friendly procedure manuals for healthcare providers, particularly those in mid-level cadres, ensures the correct use of devices [23].

In many limited resource settings, proper preparation of healthcare providers to use digital innovations for healthcare is limited [38]. To address this challenge in Tanzania, it is prudent to include other methods of intrapartum fetal surveillance in the formal training in addition to partograph and intermittent monitoring. To formalize this, stakeholders in college and higher education are required to be involved in these contents to be included in curricula to prepare the healthcare workforce for the future of digital healthcare.

Our study's findings indicated that the primary healthcare facilities lacked ambulances and reliable communication channels for sharing patient information between the referring and recipient facilities, which could be barriers to continuity of care. This is consistent with other studies conducted in Tanzania and Iran on barriers to referral systems in healthcare provision [39, 40]. A well-functioning referral system that provides continuity of care across different levels of healthcare and a reliable healthcare system with an adequate number of skilled staff are needed to increase access to maternal and child health services. The iCTG is used to screen for fetal condition; if abnormal fetal heart rate is found, the woman may need to be referred to

a higher-level facility for specialized care. However, often, there is a delay in transferring patients who need urgent care to the next-level healthcare facility. The possible explanation for this is that, in Tanzania, the referral regulations require only medical doctors to authorize the patient's referral, and the ambulance must come from the district hospital down to the health center to pick up the patient [39]. There is a need to improve the referral system in Tanzania to reduce delays contributing to increased mortality.

Our study findings show that inadequate knowledge about prenatal care and other maternal health services, including default prenatal checkups, contributes to fewer ANC visits and potentially would hinder women's acess to FHR monitoring with iCTG. In addition, disadvantaged ethnic groups are misinformed about obstetric complications due to myths and beliefs, which prevent them from receiving maternal care services in health facilities out of fear of referrals and Caesarean sections. A systematic review that was done to analyze the barriers to accessing maternal care in low-income countries in Africa highlighted that the traditions and beliefs made women hesitant to attend ANC visits in health facilities [41]. Similar findings were observed whereby women avoided using healthcare facilities because they believed institutional deliveries were only appropriate for complicated pregnancies [42, 43]. A possible solution to this barrier could be community engagement and educating pregnant women about the benefits of maternal healthcare that may increase accessibility, considering the influence of culture, beliefs, and environment on the utilization of maternity services [44, 45].

Moreover, opportunities for improving the knowledge and skills of healthcare providers, as reflected in the need for training, are a vital component of implementing iCTG. The availability of support systems for providing on-the-job training and mentorship to address the competence gap is a good indicator that healthcare providers are ready and desire to adopt new practices. In addition to training, the developed smartphone app containing reference materials for midwives and doctors on how to use iCTG and how to interpret the CTG provides an opportunity to acquire more knowledge and skills on using iCTG. Therefore, healthcare providers' readiness should be an essential starting point in building staff capacity to implement iCTG. This is consistent with what has been reported from other studies conducted in low-income countries, which state that a well-trained healthcare workforce with improved morale is crucial for the successful implementation and use of digital innovations [33, 46, 47]. This is because many developing countries have limited infrastructure and human resource capacity to improve healthcare quality.

An adequate number of women utilizing maternity care services is an excellent turn to effectively implement the iCTG and evaluate its impact on pregnancy and childbirth outcomes. In Tanzania, the percentage of pregnant women who attended more than 4 ANC visits has slightly increased from 51% in 2015 to 64.7% in 2022 [5], indicating a rise in community and women's trust in the healthcare system. However, there are still many women who are more likely to experience late detection of pregnancy-related complications, which contributes to the high maternal mortality and stillbirth rates in the country [48, 49]. As revealed in our study, the existence of CHWs for sensitization of health-seeking behaviour in the community may increase the number of pregnant women accessing care in health facilities. Moreover, our study indicated that women go to healthcare facilities primarily to learn about their condition and that of their fetus. This will serve as an opportunity for them to access monitoring of FHR with iCTG, which will reassure them about the condition of their fetus. Therefore, women will eventually enjoy the benefits of good perinatal healthcare from the hospitals utilizing iCTG.

### 4.1 Strengths and limitations of this study

This study describes the barriers and facilitators for implementing iCTG in primary healthcare facilities. The facilitators provide opportunities to generate solutions to the identified barriers

and set up context-specific implementation plans to ensure a successful strategy implementation. The qualitative approach using IDIs and FGDs for data collection ensured the richness and credibility of the findings. Nonetheless, some limitations of the present study should be acknowledged. Although the iCTG device can monitor fetal heart rate and contractions, in this phase, we mainly focused on implementing iCTG to monitor fetal heart rate during late pregnancy. Nevertheless, the later phase will assess the use of iCTG in the labour room and incorporate the component of contractions. Moreover, this analysis was limited to the qualitative aspect of the exploratory study and thus did not provide the magnitude of the barriers and facilitators revealed. However, the triangulation of study participants, settings, and researchers with different professions and experiences gives this study the strength to reflect iCTG implementation in primary healthcare facilities. Further, social desirability may provide some limitations to our findings since a midwife researcher was part of the data collectors, and HCPs may have felt obliged to provide desired responses rather than truthful ones. However, the fact that the other data collector was not an HCP, that both data collectors had adequate probing abilities, and that they were not very senior compared to the participants offset this limitation.

## 5. Conclusion

Implementing iCTG in primary healthcare facilities is affected by several barriers related to healthcare providers' competencies, referral infrastructures, human and non-human resources for healthcare, and community myths and misconceptions about maternity healthcare services. Nevertheless, the readiness of healthcare providers, available support systems, opportunities to provide on-the-job training, supportive supervision and mentorship, and community-based health programs in promoting antenatal care attendance among pregnant women are essential facilitators for implementing iCTG. We recommend capitalizing on the identified facilitators and investing to address the revealed barriers through context-specific interventions to promote fetal heart monitoring with iCTG in Tanzania and other countries of similar contexts.

## Supporting information

**S1 Table. Analysis codebook for barriers and facilitators of fetal heart monitoring with iCTG.**
(DOCX)

**S1 Checklist. Consolidated criteria for reporting qualitative studies (COREQ): 32-item checklist.**
(DOCX)

**S1 File.**
(DOCX)

**S2 File. Inclusivity in global research.**
(DOCX)

## Acknowledgments

The authors sincerely thank all participants for giving up their time to participate in this study. We also thank the hospital administration for their support during data collection. We thank Murshid Munzur E., Keiji Ninomiya, Tatsuya, Kambara, Chika Yoshikawa, and Yasuo Kawabata for participating in the project design and data collection.

## Author Contributions

**Conceptualization:** Dorkasi L. Mwakawanga, Sanmei Chen, Yhuko Ogata, Minami Suzuki, Yuryon Kobayashi, Naoki Hirose, Yoko Shimpuku.

**Data curation:** Dorkasi L. Mwakawanga, Minami Suzuki, Yuryon Kobayashi.

**Formal analysis:** Dorkasi L. Mwakawanga, Miyuki Toda.

**Funding acquisition:** Yoko Shimpuku.

**Methodology:** Dorkasi L. Mwakawanga, Yoko Shimpuku.

**Resources:** Yhuko Ogata.

**Supervision:** Sanmei Chen, Yoko Shimpuku.

**Writing – original draft:** Dorkasi L. Mwakawanga.

**Writing – review & editing:** Sanmei Chen, Yhuko Ogata, Minami Suzuki, Yuryon Kobayashi, Miyuki Toda, Naoki Hirose, Yoko Shimpuku.

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
