## [Decision Letter · Decision Letter 0]

20 May 2024

PONE-D-23-36375Barriers and facilitators for implementing fetal heart rate monitoring with a mobile cardiotocogram device (iCTG) in underserved settings: An exploratory qualitative study from TanzaniaPLOS ONE

Dear Dr. Shimpuku,

Thank you for submitting your manuscript to PLOS ONE. After careful consideration, we feel that it has merit but does not fully meet PLOS ONE’s publication criteria as it currently stands. Therefore, we invite you to submit a revised version of the manuscript that addresses the points raised during the review process.

We look forward to receiving your revised manuscript.

Kind regards,

Abera Mersha, MSc.

Academic Editor

PLOS ONE

Journal Requirements:

   "This study was funded by Japan Agency for Medical Research and Development (AMED) under Grant Number JP22hk030212."

5. In the online submission form you indicate that your data is not available for proprietary reasons and have provided a contact point for accessing this data. Please note that your current contact point is a co-author on this manuscript. According to our Data Policy, the contact point must not be an author on the manuscript and must be an institutional contact, ideally not an individual. Please revise your data statement to a non-author institutional point of contact, such as a data access or ethics committee, and send this to us via return email. Please also include contact information for the third party organization, and please include the full citation of where the data can be found.

Reviewers' comments:

Reviewer's Responses to Questions

**Comments to the Author**

1. Is the manuscript technically sound, and do the data support the conclusions?

Reviewer #1: No

Reviewer #2: Yes

Reviewer #3: Yes

2. Has the statistical analysis been performed appropriately and rigorously? 

Reviewer #1: No

Reviewer #2: Yes

Reviewer #3: I Don't Know

3. Have the authors made all data underlying the findings in their manuscript fully available?

Reviewer #1: No

Reviewer #2: Yes

Reviewer #3: Yes

4. Is the manuscript presented in an intelligible fashion and written in standard English?

Reviewer #1: Yes

Reviewer #2: Yes

Reviewer #3: Yes

5. Review Comments to the Author

Reviewer #1: 1- Lack of Novelty.

2- It seems that the researchers could not justify why a qualitative study is needed.

3- One of the strengths of this study could be providing a solution, which is not present in the results.

4- In the results section, it is not clear how many codes, initial categories and sub categories have been obtained.

5- It seems that the results do not add anything new to science. We are faced with the realities of using this device. Health service providers seem to know these facts.

Reviewer #2: 1. page 5 paragraph 2 live 8 should read thus "Despite both district hospital..................... , the 2 hospitals functions as referral

2. How many days did the collection of information take. The recording for the health workers lasted 45 to 60 mins each

3. Was the questionnaire for the health workers administered by a single researcher or more than one

4. The recording of the interview is a positive development and making of transcript by more than one person gives credence to information

Reviewer #3: Dear Authors,

I suggest the following recommendations:

1) what is the cost of the iCTG device and how do you think this could be sustainable in Tanzania.

2) how could it be useful for women considering the limiting factors you found "individual related that include inadequate knowledge and skills to use iCTG, institutional related barriers attributed to limited referral

infrastructures and shortage of staff, and community related barriers such as myths and

misconceptions that limited the number of antenatal care checkups"

3) how could it be a primary choice to invest in Tanzania, rather than increase facilities and remove barriers to access primary care rooms.

6. PLOS authors have the option to publish the peer review history of their article (what does this mean?). If published, this will include your full peer review and any attached files.

Reviewer #1: No

Reviewer #2: No

Reviewer #3: No

---

## [Author Response · Author response to Decision Letter 0]

16 Jul 2024

Point by Point Responses to the Reviewer’s Comments

Reviewer 1

1. Lack of Novelty.

Response: Thank you for this comment. We might not be very clear about the novelty of this study. To make it clearer, we improved the writing to highlight the literature gap and elaborated the novelty of this study by indicating the need to understand the barriers and facilitators to implement the FHR monitoring innovations in the undeserved countries such as Tanzania in the introduction section. In addition, we show that this is among the few qualitative studies that explored the barriers and facilitators of implementing a mobile cardiotocograph device in primary care settings in Tanzania in the discussion section. (Page 4-5, Lines 96-111; Page 18, Lines 428-430).

2. It seems that the researchers could not justify why a qualitative study is needed.

Response: Thank you for raising this concern. Qualitative study plays a crucial role in decision-making before implementing a strategy. They provide in-depth insights into different perspectives of stakeholders, which quantitative data alone may not capture. It helps to identify potential barriers, facilitators, understand contextual factors, and uncover hidden needs, ultimately leading to more informed and effective implementation strategies. We have added the justification of why exploratory qualitative study was needed before we introduce the use of iCTG in health facilities (Page 5, Lines 112-126)

3. One of the strengths of this study could be providing a solution, which is not present in the results.

Response: Thank you for this comment. The study provides insights on the things to be considered before implementation of the strategy in this context leveraging to the identified facilitators. The need and readiness to receive training and use iCTG and the available system to provide on-the job training could be used as possible solutions to fill the knowledge and skills gap. Furthermore, the availability of community health workers and the community trust to receiving ANC services are possible solutions to raising awareness and correcting myths through delivery of health education. To be clearer to the readers, we added this information in the discussion of the specified results mentioned above and in the study strengths section. (Page 19, Lines 448-450; Page 20, Lines 484-487; Page 21, Lines 492-496)

4. In the results section, it is not clear how many codes, initial categories and subcategories have been obtained.

Response: Thank you for this observation. In this study, we used thematic analysis. Braun and Clarke (2006) thematic approach analyse the underlying patterns (themes and meanings) from the shared experiences, offering a richer and more nuanced understanding of the context and the participants’ perspectives. Therefore, we presented the codes, sub-themes and themes generated during analysis. We have indicated the number of codes, subthemes and themes in the paragraph describing the themes emerged during analysis. (Page 9-10, Lines 215-235; Page 12, Lines 268-270)

5. It seems that the results do not add anything new to science. We are faced with the realities of using this device. Health service providers seem to know these facts. 

Response: Thank you for raising this concern. This study aimed to analyse the barriers and facilitators on the implementation of iCTG specifically on primary healthcare facilities to inform the development of intervention and training for healthcare providers. The results presented in this paper add merit to the science in the fact that, we describe why we needed evidence for implementation barriers and facilitators to determine strategies for optimal use of iCTG in limited resource settings. In addition, by reading the findings of this paper, 

future implementors would benefit by identifying the available context specific opportunities/strategies to support the implementation of evidence-based interventions in the healthcare system despite the faced challenges/realities. (Page 12-18, Lines 267-423)

Reviewer 2

1. Page 5 paragraph 2 live 8 should read thus "Despite both district hospital..................... , the 2 hospitals functions as referral Response: Thank you for the suggestion. We have revised the sentence. Now reads as “Despite both the district hospital and the health center falling within the primary level of the pyramidal Tanzanian health service delivery system, the district hospital functions as a referral hospital for their immediate health facilities due to their greater capacity to provide comprehensive emergency obstetrics and newborn care.” (Page 6, Lines 134-137)

2. How many days did the collection of information take. The recording for the health workers lasted 45 to 60 mins each

Response: Thank you for this comment. We used a total of 25 days for data collection; however, we spent more days in the file for other procedures including contacting hospital administrators, recruitment and setting up appointment for interviews and FGDs. 

We revised the data collection section to include the information on the number of days spent for data collection and edited the last sentence about the time used to conduct interviews. Now reads as “The interviews lasted between 45 and 60 minutes”. (Page 8, Lines 189-192; Page 9, Lines 202-203)

3. Was the questionnaire for the health workers administered by a single researcher or more than one. 

Response: Thank you for raising this concern. The data collection team was composed of five people (3 researchers and 2 research assistants). Each session of interview or FGD was conducted by a pair of two people. One asking questions and another taking notes and recording the session. At this moment, other people waited outside the room and joined the interviewing team during the debriefing to hear what was transpired during the interview/FGD. Then, the following interview was done by a different team. We included this information in the methods section. (Page 8, Lines 184-194)

4. The recording of the interview is a positive development and making of transcript by more than one person gives credence to information. 

Response: Thank you for this information which complement to what we observed. We audio-recorded all interview and discussions to ensure that all participant’s accounts are captured. The transcripts produced were made from more than one person. The maximum use of participants’ accounts and the use of more than one data collector ensured the credibility of the data. We included this information in the methods section. (Page 11, Lines 244-247)

Reviewer 3

1. What is the cost of the iCTG device and how do you think this could be sustainable in Tanzania. 

Response: Thank you for raising this concern. There are various types of iCTG based on their functions. For example, the common type of iCTG has three primary functions: (1) monitoring FHR; (2) monitoring contractions; and (3) sharing data via the internet. So, if the context-specific requirement does not require the function of sharing data due to any reason or due to internet challenges, then iCTG can be customized with only two major functions: monitoring fetal heart rate and contractions. And this makes the purchase price lower compared to the original price. The one discussed in this paper has all three functions and costs about 4.5 million Tanzanian shillings. However, since the company is currently working on reducing the price, we did not include the information in the manuscript. If our study shows positive health impacts in Tanzania, the results will provide strong evidence and valuable lessons learned for sustainability of this innovative healthcare technologies to save lives, impact the quality of care, and enable women experience a positive childbirth experience.

2. How could it be useful for women considering the limiting factors you found "individual related that include inadequate knowledge and skills to use iCTG, institutional related barriers attributed to limited referral

infrastructures and shortage of staff, and community related barriers such as myths and

misconceptions that limited the number of antenatal care checkups" 

Response: Thank you for asking. The identified barriers and facilitators will help the researchers to design the intervention and prepare for implementation. This included: the provision of training to the healthcare providers to fill out the gap of knowledge and skills to use the iCTG. To raise awareness among consumers, whereby community health workers will be used to sensitize women to attend antenatal care. But also, we developed a smartphone app with educative contents for midwives to use during the provision of health education to pregnant women as well as informing them about the benefits of having regular antenatal checkups. Therefore, women will eventually enjoy the benefits of good perinatal healthcare from the health facilities utilizing iCTG. We included this point in the discussion. (Page 21-22, Lines 488-514)

3. How could it be a primary choice to invest in Tanzania, rather than increase facilities and remove barriers to access primary care rooms. 

Response: Thank you for your concern. We acknowledge that the use of iCTG may be the primary choice to invest at least at the moment in Tanzania. This is because, improving the situation is not limited to increasing facilities and removing barriers to access primary care. Tanzania is facing a critical shortage of human resource for health; this necessitates the implementation of suitable technologies/innovations for healthcare delivery as one of the solutions to this problem as well to improving the health outcomes. 

Reducing maternal and neonatal mortality is still a priority agenda in Tanzania. Currently, the neonatal mortality is at 24 deaths per 1,000 live births, which is much higher than the one indicated in SDG 3. Intensive fetal monitoring during late pregnancy and in labour plays an important role in detecting perinatal hypoxia early and saving lives. The iCTG portable device has been recommended as a suitable strategy to ensure perinatal safety in settings with medical resource challenges including shortage of staff. We included this information in the introduction. (Page 3, Lines 65-70; Page 4, Lines 84-92)

---

## [Decision Letter · Decision Letter 1]

14 Oct 2024

PONE-D-23-36375R1**Barriers and facilitators of fetal heart monitoring with a mobile cardiotocogram (iCTG) device in underserved settings: An exploratory qualitative study from Tanzania**PLOS ONE

Dear Dr. Shimpuku,

Thank you for submitting your manuscript to PLOS ONE. PLOS One acknowledge the process of your endeavor in revising your manuscript to meet its publication criteria. After careful consideration, we feel that it has merit but still it requires further corrections and revisions based on the reviewers recent comments to fully meet PLOS ONE’s publication criteria. Therefore, we invite you to submit a revised version of the manuscript that addresses the points raised during the review process.

**The points of focus by the academic editor: **The conclusions must be drawn appropriately based on the data presented.The language edition of the manuscript should be made by native English speaker.Make sure to address all comments forwarded by reviewers and incorporate the revised changes.Provide specific feedback from your evaluation of the manuscript through point by point response.In your manuscript you give main focus only on heart rate, with no mention of uterine contractions throughout the paper seems a limitation, please check if you can incorporate it briefly on the limitation section.==============================

We look forward to receiving your revised manuscript.

Kind regards,

Tesfay Gebregzabher Gebrehiwet, PhD

Academic Editor

PLOS ONE

Journal Requirements:

Reviewers' comments:

Reviewer's Responses to Questions

**Comments to the Author**

1. If the authors have adequately addressed your comments raised in a previous round of review and you feel that this manuscript is now acceptable for publication, you may indicate that here to bypass the “Comments to the Author” section, enter your conflict of interest statement in the “Confidential to Editor” section, and submit your "Accept" recommendation.

Reviewer #4: (No Response)

Reviewer #5: (No Response)

2. Is the manuscript technically sound, and do the data support the conclusions?

Reviewer #4: Partly

Reviewer #5: Partly

3. Has the statistical analysis been performed appropriately and rigorously? 

Reviewer #4: Yes

Reviewer #5: Yes

4. Have the authors made all data underlying the findings in their manuscript fully available?

Reviewer #4: Yes

Reviewer #5: Yes

5. Is the manuscript presented in an intelligible fashion and written in standard English?

Reviewer #4: Yes

Reviewer #5: No

6. Review Comments to the Author

Reviewer #4: Comments

Overall, the response given to prior reviewers addresses the majority of the raised issues. I have some comments and suggestions to be addressed which are mentioned below.

The whole paper needs a grammar edition

This is a good entry point for identifying the obstacles before any implementation research is done so that some problems may be fixed beforehand and correctly. But there are a lot of barriers identified and it is worth mentioning the solutions in summary or table or figure which are listed by the participants and may help in addressing the root cause during the implementation.

Page 3, Line 70-72, I think this is a vague meaning and no clear meaning of mentioning this in this paper's context or must be with a reason why not effective if there is access to them? What makes the difference in safety, appropriateness, WHY?

Issue of feasibility and sustainability iCTG use in Tanzania?

Shortage of staff is one of your barriers and you mentioned it as a solution for the use of iCTG. Does it contradict? Especially for those who receive the record and interpret and decide and intervene (GYN/OBS specialists)

Issue of Referral is a big barrier vs the use of iCTG? How are you going to negotiate this?

Reviewer #5: Thank you for letting me review your paper. The research question is interesting to the Journal’s audiences.

There are multiple typos, spelling errors, grammatical errors, poor word usage, punctuation, and formatting problems. These must be addressed to fulfill the standards of the journal. The language should undergo proofreading by a native or get professional English editing services.

To mention a few … lines 85, 86, 89, 90, 96,103, 144, 147, 233 typo errors, punctuation….

The AI score of the study is 5%, which is commendable

The plagiarism score is 13%, which indicates further rephrasing is required to keep it below 10%

CTG stands for cardiotocography, which involves monitoring the heart rate (cardio) and uterine contractions (toco). However, the manuscript seems to focus only on heart rate, with no mention of uterine contractions throughout the paper (It was mentioned only once, line 88). It is unclear to refer to it as CTG when it is merely Doppler due to this error. It is imperative that this limitation be addressed in the section on limitations of the study.

What was the professional level of the individual who conducted IDI? How was bias mitigated in the selection, of the women, as you mentioned that the on-call midwife was in charge of selecting and organizing the pregnant mothers? Could there be a bias to choose inherently pregnant mothers who are satisfied with the overall care including the CTG? Who selected the interviewees, if the midwife was only, limited to selecting a private place only?

The FGDs were conducted by the investigators, DLM, and co. What measures were taken to decrease bias? Was there any bracketing? it needs to be mentioned in the methods section. Other than your statement ‘’generated inductively using thematic analysis and presented with the support of subthemes and succinct quotes’’

Institutional Barriers

I am having trouble relating these findings, especially about the study's objective. Are you trying to say that health professionals do not use CTG because anomalies in the referral system prevent them from sending patients when they detect fetal heart abnormalities?

Community-related barriers,

The FGD should have focused on i CTG instead of exploring the whole health system. Is the long queue due to the usage of i CTG? You need to clearly address it in the discussion section.

7. PLOS authors have the option to publish the peer review history of their article (what does this mean?). If published, this will include your full peer review and any attached files.

Reviewer #4: No

Reviewer #5: **Yes: **Awol Yemane Legesse

---

## [Author Response · Author response to Decision Letter 1]

5 Nov 2024

Academic editor comments

1. The conclusions must be drawn appropriately based on the data presented.

Response: Thank you for the comment. We revised the conclusion for both the abstract and the main paper. Now, it corresponds with what is presented in the data/results. (Page 2-3, Lines 51-58)

2. The language edition of the manuscript should be made by native English speaker.

Response: This is well noted. The language editing has been done to correct grammar errors by a native English speaker.

3. In your manuscript you give main focus only on heart rate, with no mention of uterine contractions throughout the paper seems a limitation, please check if you can incorporate it briefly on the limitation section.

Response: Thank you for raising this concern. In this phase, we mainly focused on the use iCTG to monitor fetal heart rate during late pregnancy to improve pregnancy and childbirth outcomes. The later phase will assess the use of iCTG in the labour room and incorporate the component of contractions. We have provided a brief explanation in the limitation section. (Page 23, Lines 523-526)

Reviewer 4

1. The whole paper needs a grammar edition.

Response: Thank you for your concern. We have gone through the manuscript and corrected grammatical errors accordingly.

2. This is a good entry point for identifying the obstacles before any implementation research is done so that some problems may be fixed beforehand and correctly. But there are a lot of barriers identified and it is worth mentioning the solutions in summary or table or figure which are listed by the participants and may help in addressing the root cause during the implementation.

Response: Thank you for raising this concern. We have provided a table (Table 2) in the results section that shows the potential solutions reported by the participants. (Page 18-19, Lines 426-427)

3. Page 3, Line 70-72, I think this is a vague meaning and no clear meaning of mentioning this in this paper's context or must be with a reason why not effective if there is access to them? What makes the difference in safety, appropriateness, WHY?

Response: Thank you for this observation. We have revised the sentence. Now reads as ‘Many medical devices designed for use in high-income countries have several limitations when reciprocated in low-resource settings. [8, 9]. Currently, technological innovations are being leveraged to ensure the availability of suitable and appropriate devices for monitoring fetal surveillance in LMICs [8]’. (Page 3, Lines 70-73).

4. Issue of feasibility and sustainability iCTG use in Tanzania?

Response: Thank you for raising this concern. Our feasibility check indicated that the use of iCTG in Tanzania was feasible, needed and acceptable. The sustainability will depend on our results after the implementation. If our study shows positive health impacts in Tanzania, the results will provide strong evidence and valuable lessons learned for sustainability of this innovative healthcare technologies to save lives, impact the quality of care, and enable women experience a positive childbirth experience.

5. Shortage of staff is one of your barriers and you mentioned it as a solution for the use of iCTG. Does it contradict? Especially for those who receive the record and interpret and decide and intervene (GYN/OBS specialists)

Response: Thank you for raising this concern. One innovative part of iCTG is that it doesn’t require continuous provider attendance during measurement compared to an intermittent fetal monitor which requires constant provider attendance to a patient. Thus, it is considered suitable in settings with resource challenges, including a shortage of staff.

We acknowledge that our findings contradict what we report as a solution. However, participants (providers) perceived the staff shortage as a barrier to using iCTG because they considered a 20-minute measurement amidst a long queue of pregnant women waiting to receive service from a single provider a problem. Moreover, there was a limited number of OBGY specialists to receive the recordings, interpret them, and make decisions. 

6. Issue of Referral is a big barrier vs the use of iCTG? How are you going to negotiate this?

Response: Thank you very much for this concern. Ambulances were stationed in zones/areas with referral issues, which helped to overcome the referral barrier. The authors have included a table of the proposed solutions to the identified barriers in the results section. (Page 18-19, Lines 426-427)

Reviewer 5

1. There are multiple typos, spelling errors, grammatical errors, poor word usage, punctuation, and formatting problems. These must be addressed to fulfill the standards of the journal. The language should undergo proofreading by a native or get professional English editing services.

To mention a few … lines 85, 86, 89, 90, 96,103, 144, 147, 233 typo errors, punctuation….

Response: Thank you for this valuable comment. The language editing has been done to correct grammar errors by a native English speaker.

2. The AI score of the study is 5%, which is commendable

The plagiarism score is 13%, which indicates further rephrasing is required to keep it below 10%

Response: Thank you for this valuable comment. We have done further rephrasing which might reduce the percentage of plagiarism.

3. CTG stands for cardiotocography, which involves monitoring the heart rate (cardio) and uterine contractions (toco). However, the manuscript seems to focus only on heart rate, with no mention of uterine contractions throughout the paper (It was mentioned only once, line 88). It is unclear to refer to it as CTG when it is merely Doppler due to this error. It is imperative that this limitation be addressed in the section on limitations of the study.

Response: Thank you for raising this concern. The iCTG device can monitor both fetal heart rate and contractions, however, this study only focused on implementing iCTG during late pregnancy to improve fetal outcomes. Nevertheless, the later phase of the study focused on the use of iCTG during labour and incorporated the components of monitoring fetal heart rate and contractions. We have addressed this concern in the limitation section. (Page 23, Lines 523-526)

4. What was the professional level of the individual who conducted IDI? How was bias mitigated in the selection, of the women, as you mentioned that the on-call midwife was in charge of selecting and organizing the pregnant mothers? Could there be a bias to choose inherently pregnant mothers who are satisfied with the overall care including the CTG? Who selected the interviewees, if the midwife was only, limited to selecting a private place only?

Response: Thank you very much for asking. Regarding the individuals who conducted IDI; one had a bachelor’s degree, and two had a master’s degree education level. (Page 8, Lines 186-188)

The midwife was not involved in the selection of study participants. Rather, she introduced the researcher to women who were available at the clinic. The selection of the eligible participants was solely conducted by the data collection team. (Page 7, Lines 154-158)

5. The FGDs were conducted by the investigators, DLM, and co. What measures were taken to decrease bias? Was there any bracketing? it needs to be mentioned in the methods section. Other than your statement ‘’generated inductively using thematic analysis and presented with the support of subthemes and succinct quotes’’

Response: Thank you for asking. We relied on the following methods to reduce bias: (Page 11, Lines 245-256)

a) Triangulation of researchers. We had more than one interviewers.

b) Triangulation of study participants. We collected data from both supply and demand side. And we ensured maximum variation among study participants. 

c) Triangulation of data coders. We employed a team-based approach for coding. 

d) Checking for coding reliability. At first, we both coded the same transcript, and then we convened to resolve the discrepancies and disagreements that emerged after coding same transcript before we continued with other transcripts. 

e) Lastly, researchers not being seniors or early career made participants to be more comfortable.

6. Institutional Barriers

I am having trouble relating these findings, especially about the study's objective. Are you trying to say that health professionals do not use CTG because anomalies in the referral system prevent them from sending patients when they detect fetal heart abnormalities?

Response: Thank you for raising this concern. This qualitative study was done to identify potential barriers, facilitators, understand contextual factors, and uncover hidden needs, that could lead to more informed and effective implementation strategies. Identifying the obstacles helped in designing intervention and fixing some problems beforehand. (Page 5, Lines 114-125)

Our findings indicated that, the referral system was a barrier in case there was a need for referral after the detection of abnormal FHR with iCTG. Nevertheless, this was not a barrier for healthcare providers not to use the iCTG rather, it was foreseen as a barrier only in the need of a referral following the results of iCTG recording. Therefore, the potential solution to this obstacle was to station the ambulances in the zones/areas which had referral system challenges. (Page 18-19, Lines 426-427)

7. Community-related barriers,

The FGD should have focused on iCTG instead of exploring the whole health system. Is the long queue due to the usage of iCTG? You need to clearly address it in the discussion section.

Response: Thank you for this comment. Before the implementation of iCTG we wanted to explore the need from the demand side on readiness towards utilization of the service. Thus, we went further to interviewing pregnant women. From the interviews, it came out that illiteracy regarding ANC checkup might affect the utilization of the service. Some women reported defaulting to follow-up checkups as a result of the long queue for antenatal care services in healthcare facilities. (Page 21, Lines 478-490)

---

## [Editor Report · Decision Letter 2]

18 Nov 2024

Barriers and facilitators of fetal heart monitoring with a mobile cardiotocograph  (iCTG) device in underserved settings: An exploratory qualitative study from Tanzania

PONE-D-23-36375R2

Dear Dr. Yoko Shimpuku

We’re pleased to inform you that your manuscript has been judged scientifically suitable for publication and will be formally accepted for publication once it meets all outstanding technical requirements.

Kind regards,

Tesfay Gebregzabher Gebrehiwet, PhD

Academic Editor

PLOS ONE